# Assessing CYP2C8-Mediated Pharmaceutical Excipient-Drug Interaction Potential: A Case Study of Tween 80 and Cremophor EL−35

**DOI:** 10.3390/pharmaceutics13091492

**Published:** 2021-09-17

**Authors:** Chengming Wen, Haoyang Hu, Wenwen Zhang, Xin Liu, Xuehua Jiang, Ling Wang

**Affiliations:** Key Laboratory of Drug-Targeting and Drug Delivery System of the Education Ministry, Department of Clinical Pharmacy and Pharmacy Administration, West China School of Pharmacy, Sichuan University, Chengdu 610041, China; chengmingscu@163.com (C.W.); hhy15923016890@163.com (H.H.); Zhang17882085718@163.com (W.Z.); liuxx132@126.com (X.L.); jxh1013@scu.edu.cn (X.J.)

**Keywords:** pharmaceutical excipients, CYP2C8, Cyp2c22, Tween 80, EL−35, enzyme inhibitory, pharmaceutical excipients-drug interaction

## Abstract

Pharmaceutical excipients (PEs) are substances included in drug formulations. Recent studies have revealed that some PEs can affect the activity of metabolic enzymes and drug transporters; however, the effects of PEs on CYP2C8 and its interaction potential with drugs remain unclear. In this study, we evaluated the effects of Tween 80 and EL−35 on CYP2C8 in vitro and further investigated their impacts on the PK of paclitaxel (PTX) in rats after single or multiple doses. The in vitro study indicated that Tween 80 and EL−35 inhibited CYP2C8 activity in human and rat liver microsomes. EL−35 also decreased the expression of CYP2C8 in HepG2 cells. In the in vivo study, Tween 80 did not alter the PK of PTX after single or multiple doses, whereas EL−35 administered for 14 days significantly increased the AUC and MRT of PTX. Further analysis indicated that multiple-dose EL−35 reduced the expression of Cyp2c22 and production of 6-OH-PTX in the rat liver. Our study suggested that short-term exposure to both PEs did not affect the PK of PTX in rats, but multiple doses of EL−35 increased the AUC and MRT of PTX by downregulating the hepatic expression of Cyp2c22. Such effects should be taken into consideration during drug formulation and administration.

## 1. Introduction

Pharmaceutical dosage formulations usually contain both pharmacologically active compounds and excipients to produce proper formulations for patients [1,2,3]. Although most pharmaceutical excipients (PEs) are inactive, they are critical and essential components in finished pharmaceutical products, and they can be used as binders, disintegrants, and surfactants, etc. [4]. For example, surfactants are used to solubilize hydrophobic drugs, methylcellulose can be used to prepare drug suspensions or added to tablets as a disintegrating agent, and cyclodextrin can be used to improve drug stability or control drug release [5]. However, not all PEs are “inactive”, and some are reported to affect the activity of metabolic enzymes, such as cytochrome P450 (CYP450) 3A4/5 (CYP3A4/5), CYP2C9, CYP2C19, CYP2D6, CYP2E1, CYP1A2, and UGT1A1 [6,7,8,9,10], or drug transporters, such as P-gp, BCRP, MRP2, and OATP1A2/2B1 [11,12,13,14]. For example, Martin and colleagues investigated the impact of 23 commonly used excipients (10 polymers and 13 surfactants) on CYP2E1, CYP3A4, CYP3A5, CYP2C9, CYP2C19, CYP1A2, and CYP2D6 using baculosome-derived CYP450 enzymes across a range of concentrations [10]. The investigators found that most excipients were capable of inhibiting or increasing the activity of several different CYP450 isoforms. Furthermore, the effects of PEs were exerted on both drug metabolism and absorption [15]. Zhang et al. reviewed the effects of PEs on gastrointestinal tract metabolic enzymes and drug transporters, observing that more than 60 excipients interfered with metabolic enzymes or transporters [9]. These effects may further influence the pharmacokinetics (PK) of active components, leading to changes in their pharmacodynamics in the body.

Tween 80 and EL−35, also named Polysorbate 80 and Polyoxyl 35 hydrogenated castor oil, respectively, are common excipients and solubilizing agents used in the pharmaceutical industry. These PEs were reported to inhibit the activity of CYP3A4 in human liver microsomes (HLMs) with IC_50_s of 0.4 and 0.6 mM, respectively [7]. However, the effects of Tween 80 and EL-35 on the activity or expression of CYP2C8 have not been revealed.

CYP2C8 is an essential member of the CYP2C family, and it is mainly abundant in the human liver and present at lower levels in the intestine and kidneys. CYP2C8 accounts for approximately 6–7% of hepatic CYP450s and metabolizes more than 100 drugs, including paclitaxel (PTX), rosiglitazone, pioglitazone, repaglinide, imatinib, cerivastatin, enzalutamide, and atorvastatin [16,17,18,19,20,21]. Recent studies demonstrated that CYP2C8 participates in the synthesis of epoxyeicosatrienoic acids and reactive oxygen species, which have vasodilative, anti-inflammatory, and anti-atherosclerosis effects in the human cardiovascular system [22].

In rats, the Cyp2c family includes several isoforms, such as Cyp2c6, Cyp2c7, Cyp2c11, Cyp2c12, Cyp2c13, Cyp2c22, and Cyp2c23. The Cyp2c12 and Cyp2c7 isoforms are female-predominant; in contrast, Cyp2c11 is the predominant isoform in male rat liver, comprising up to 50% of the total Cyp450 content, and Cyp2c13 is also male-specific. Meanwhile, Cyp2c6 is expressed gender-independently. Cyp2c23 is highly expressed in rat kidneys, and it has been suggested to be important for producing compensatory renal artery vasodilatation in response to salt loading [23]. Cyp2c22 is expressed almost exclusively in hepatocytes, making it similar to CYP2C8 in humans. According to Qian and colleagues, Cyp2c22 has 60 and 62% amino acid identity to CYP2C8 and CYP2C9, respectively, and 80% sequence similarity to both human proteins, indicating that Cyp2c22 is homologous to human CYP2C8 and CYP2C9 [24].

PTX is used to treat several types of cancer, including ovarian cancer, breast cancer, lung cancer, Kaposi sarcoma, cervical cancer, and pancreatic cancer [25]. It is metabolized primarily to 6α-OH-PTX by CYP2C8 and to two minor metabolites, namely 3′-p-OH-PTX and 6α, 3′-p-dihydroxy-PTX, by CYP3A4. Taxol, the branded formulation of PTX, contains 527 mg/mL EL−35 in its formulation, reflecting a considerably large amount in a drug formulation and was reported to be a vital factor that led to the nonlinear kinetics for PTX and the two primary metabolites [26]; however, no study has evaluated the effects of EL−35 on CYP2C8 activity and the potential interaction with PTX.

Thus, in this study, we first assessed the effects of Tween 80 and EL−35 on the activity and expression of CYP2C8 in HLMs and rat liver microsomes (RLMs) and HepG2 cells. We also investigated the effects of these PEs on the pharmacokinetics of PTX following single- or multiple-dose administration in Wistar rats to evaluate the potential interaction between these two PEs and PTX.

## 2. Materials and Methods

### 2.1. Materials

Macrogol 600 (PEG 600), Cremophor EL−35 (EL−35), quercetin (QCT), carbamazepine (CBZ), PTX, β-NADP, D-glucose-6-phosphate (G-6-P), glucose-6-phosphate dehydrogenase (G-6-PDH), KH_2_PO_4_, Na_2_HPO_4_, MgCl_2_, DTT, and EDTA were purchased from Meilun Biological Technology (Dalian, China). 6-OH-PTX was purchased from Toronto Research Chemicals (Toronto, ON, Canada). Tween 80 was purchased from Well Pharmaceutical (Nanjing, China). PMSF and heparin were purchased from Sigma-Aldrich (St. Louis, MO, USA). HLMs (Mixed Gender 50-Donor Pooled) were purchased from Bioreclamation IVT (Baltimore, MD, USA).

### 2.2. Animals and Experimental Design

Male Wistar rats (in-house random-bred), aged 8–12 weeks and weighing 225–300 g, were quarantined in the animal house of the West China School of Pharmacy, Sichuan University (Chengdu, China), for 14 days under a 12 h/12 h dark/light cycle. Rats were randomly divided into six groups (*n* = 6 per group). For single-dose administration, one group administered saline served as a blank control, and the other two groups were intravenously administered a single dose of Tween 80 (180 mg/kg) or EL−35 (430 mg/kg). For multiple-dose administration, animals were intravenously administered saline, Tween 80 (180 mg/kg) or EL−35 (430 mg/kg) for 14 consecutive days. The dosages used in this study were set as 1/10 LD_50_ for both PEs according to the FDA database of inactive ingredients.

After single- or multiple-dose administration, rats were treated with 3 mg/kg PTX solution (prepared in a solvent mixture containing 61% PEG 600 (5% *w*/*v*) and 49% ethanol) via caudal vein injection. PEG 600 exerted no impact on CYP2C8 activity in HLMs and RLMs (Appendix A). Blood samples (200 μL) were collected at 6 min, 15 min, 30 min, 1 h, 2 h, 3 h, 4 h, 6 h, 8 h, 12 h, and 24 h after administration from the retro-orbital plexus into heparinized microcentrifuge tubes (approximately 20 IU heparin/mL blood). Rats were anesthetized by intraperitoneal injection of 50% urethane (3 mL/kg) after the last blood sample was collected, and livers were harvested for qPCR analysis and RLM extraction.

### 2.3. In Vitro Metabolism Study

The basic incubation medium contained 50 mM potassium phosphate buffer, pH 7.4 (KPI), a NADPH-regenerating system (1 mM NADP, 5 mM G-6-P, 1 U/mL G-6-PDH, and 5 mM MgCl_2_), 0.25 mg/mL HLMs or 1 mg/mL RLMs, and the probe substrate PTX. The final incubation volume was 100 µL, and the organic solvent concentrations were less than 1%. HLMs/RLMs were incubated with PTX at 37 °C for 1 h. At the end of the incubation, the reaction was terminated by adding 100 µL of acetonitrile containing the internal standard CBZ. After vortexing and centrifugation at 14,000 rpm for 5 min, the supernatant was analyzed by HPLC–MS/MS for CYP2C8-specific 6α′-hydroxylation of PTX. All experiments were performed in triplicate.

To determine the general kinetics of CYP2C8 in microsomes, HLMs/RLMs were incubated with 5, 10, 15, 20, 25 or 30 μM PTX for 1 h. Km and Vmax were calculated using nonlinear regression analysis by GraphPad 7.00 (Appendix A).

The effects of PEs on CYP2C8-specific PTX 6α′-hydroxylation in HLMs/RLMs were examined by adding the test PE (1 mg/mL) or vehicle (blank control) to the incubation mixture. The percent rate of control was calculated from the 6-OH-PTX production rates in the presence of the PE versus its absence (blank control). QCT (10 µM) was used as the positive control of CYP2C8 inhibition.

The IC_50_s of Tween 80 and EL-35 on CYP2C8 were determined by incubating HLMs with 10 μM PTX in the presence of a series of concentrations of PEs for 1 h at 37 °C. The concentration range was 0.03125–2 mg/mL for EL−35 and 0.0625–2.5 mg/mL for Tween 80. Activities were expressed as a percentage of the 6-OH-PTX production in the negative control. To preliminarily characterize the inhibitory type of the PEs against CYP2C8, 0.5 mg/mL EL−35 or Tween 80 was co-added to the incubation system with PTX (5, 10, 20, 25, or 30 µM). Inhibition data were plotted as a Lineweaver–Burk plot.

### 2.4. RLM Extraction

Six male Wistar rats (aged 8–12 weeks and weighing 200–250 g) were fasted overnight for 12 h before the experiment and anesthetized via an intraperitoneal injection of 50% (*w*/*v*) urethane solution (3 mL/kg). The abdominal cavity was opened along the midline of the abdomen, and an infusion of pre-cooled (4 °C) wash buffer (containing 1.09 mg/mL KH_2_PO_4_, 7.96 mg/mL Na_2_HPO_4_, 0.56 mg/mL EDTA, 0.154 mg/mL DTT, and 0.04 mg/mL PMSF) was administered from the hepatic portal vein to remove most of the blood in the liver. Then, the liver was cut into pieces, transferred to homogenization tubes, and incubated in pre-cooled (4 °C) homogenization buffer (containing 10 mM KPI, 85.6 mg/mL sucrose, and 0.373 mg/mL EDTA) for homogenization. The obtained homogenate was centrifuged at 12,000× *g* and 4 °C for 15 min. After the centrifugation, the upper layer was collected and centrifuged at 110,000× *g* and 4 °C for 1 h. The upper layer was discarded, and an appropriate volume of 250 mM sucrose solution was added to the pellet. The mixture was stirred evenly to obtain pooled RLMs. All experimental operations were performed at 4 °C.

### 2.5. Sample Analysis

The concentrations of PTX and 6-α-OH-PTX in rat plasma and the HLM/RLM incubation system were analyzed by HPLC–MS/MS. Briefly, PTX, 6-α-OH-PTX, and CBZ (internal standard) were extracted using a protein precipitation method with acetonitrile and detected by multiple reaction monitoring of the m/z transitions 854.2–285.9, 870.8–286.1, and 237–194 for PTX, 6-α-OH-PTX, and CBZ, respectively. Briefly, 50 μL plasma samples were spiked with 10 μL of internal standard solution and the mixture was vortex-mixed for 30 s. The mixtures were then precipitated with 150 μL acetonitrile by vortex-mixing for 3 min. The sample was centrifuged at 14,000 rpm for 5 min, and 10 μL of the supernatant was analyzed by the LC–MS/MS system. Chromatographic separation was performed on the mass spectrometry column (CAPCELL PAK C18, 50 × 2.00 mm, 5 µm) coupled with a Security Guard cartridge (C18, 4 × 3.0 mm i.d., Phenomenex). The gradient elution procedures are listed in Appendix A. The pharmacokinetic parameters of PTX were calculated by noncompartmental analysis using the Phoenix WinNonlin software (version 6.3, Pharsight Corp, Mountain View, CA, USA).

Serum indices of liver function, including AST/ALT/ALP, were analyzed by West China Frontier Pharmatech (Chengdu, China) using a biochemistry analyzer (ROCHE COBAS Integra 400 Plus).

### 2.6. Cell Culture and Experimental Design

HepG2 human hepatoma cells were obtained from the Chinese Academy of Sciences and cultured at 37 °C in a humidified atmosphere containing 5% CO_2_. The cells were routinely screened for mycoplasma contamination. To determine the effects of Tween 80 and EL−35 on CYP2C8 and CYP3A4 expression in HepG2 cells, we first assessed the cytotoxicity of a series of concentrations of these two PEs in HepG2 cells after 24 h of culture. Cells were treated with nontoxic concentrations in subsequent experiments. After incubation, total RNA and protein were extracted, and the mRNA and protein expression of CYP2C8 and CYP3A4 was compared with that of the blank control by RT-qPCR and Western blotting.

### 2.7. MTT Assay

MTT assays were performed to evaluate the cytotoxicity of Tween 80 and EL−35 in HepG2 cells. Briefly, HepG2 cells were seeded in a 96-well plate at a density of 4 × 10^4^ cells/well and cultured in DMEM containing 10% FBS. The following day, the culture medium was removed and 100 μL of PE solution (concentrations of 0.05, 0.1, 0.2, 0.4, 0.5, 0.6, 0.8, 0.9, or 1 mg/mL, prepared in DMEM containing 1% FBS) or blank DMEM (control) was added to each well. The cells were then incubated at 37 °C for 24 h. After removing PE solutions, 100 μL of the MTT solution (0.5 mg/mL dissolved in PBS buffer) was added to each well, and the plate was incubated for 4 h at 37 °C. After the incubation, the medium was removed and 100 μL of DMSO was added to the wells to solubilize the formazan product. A colorimetric assay was performed at 490 nm using a Multiskan MK3 Reader (Thermo Fisher Scientific, Waltham, MA, USA).

### 2.8. RT-qPCR Analysis

Total RNA extraction from HepG2 cells and rat livers was performed using TRIzol reagent (Gbcbio, China) according to the manufacturer’s instructions. RNA (1 µg) was used as a template for cDNA synthesis using Hifair™ 1st strand cDNA Synthesis SuperMix (Yeasen Biological Technology Co. Ltd. Shanghai, China). RT-qPCR was performed using Hieff™ qPCR SYBR^®^ Green Master Mix (Yeasen Biological Technology Co. Ltd. Shanghai, China) using specific primers (Appendix A). The amplification protocol consisted of initial denaturation at 95 °C for 5 min, followed by 40 cycles of denaturation at 95 °C for 10 s, annealing at 60 °C for 20 s, and extension at 72 °C for 20 s. The relative gene expression was normalized against that of human GAPDH or rat Gapdh. Gene expression was calculated using the 2^−ΔΔCT^ method. The primers were obtained from Tsingke Biological Technology (Chengdu, China).

### 2.9. Western Blot Analysis

Cells were homogenized in RIPA lysis buffer. Whole-cell extracts were prepared by direct lysis in 1× electrophoresis sample buffer. The protein content was determined using a BCA protein assay kit (Biyuntian Co Ltd., Shanghai, China). Total cellular protein was resolved by 10% SDS-PAGE and transferred onto a polyvinylidene difluoride membrane. The membrane was blocked with 5% nonfat milk and incubated with the primary antibody overnight at 4 °C, followed by incubation with the secondary antibody for 1 h. Antibodies against CYP2C8 and CYP3A4 were obtained from Proteintech Biotechnology. All antibodies were used at the dilutions recommended by the manufacturers. The densities of the protein bands were determined using ImageJ software (National Institutes of Health, Bethesda, MD, USA).

### 2.10. Statistical Analysis

Statistical analysis was performed using IBM SPSS Statistics version 22 (IBM, Armonk, NY, USA). One-way ANOVA with Bonferroni’s multiple comparison test was used to analyze most sets of quantitative data. If the data did not meet normality or homogeneity of variance, nonparametric analysis using the Kruskal–Wallis test was conducted. All other analyses were performed using Student’s *t*-test. The level of significance was set at *p* < 0.05. Data are presented as the mean ± standard deviation.

## 3. Results

### 3.1. Inhibitory Effects of Tween 80 and EL−35 on CYP2C8 Activity in HLMs/RLMs

The effects of Tween 80 and EL−35 on CYP2C8 activity in HLMs/RLMs were evaluated as described previously. QCT, which served as the positive control of CYP2C8 inhibition, inhibited PTX 6α′-hydroxylation, as reported previously. The results indicated that Tween 80 and EL−35 consistently inhibited PTX 6α′-hydroxylation in both HLMs and RLMs (Appendix A). Furthermore, we determined the IC_50_s of the two PEs in HLMs. The IC_50_s of Tween 80 and EL-35 were 1.447 and 1.042 mg/mL, respectively (Figure 1A,B). To preliminarily characterize the inhibitory types of these two against CYP2C8, EL−35 and Tween 80 were co-added at a concentration of 0.5 mg/mL to the incubation system with PTX. Inhibition data are plotted as a Lineweaver–Burk plot in the presence and absence of PEs (Figure 1C,D). Based on the results, we found that Tween 80 and EL−35 did not match the three classical inhibition types.

### 3.2. Effects of Tween 80 and EL−35 on CYP2C8 and CYP3A4 Expression in HepG2 Cells

The effects of Tween 80 and EL−35 on the expression of human CYP2C8 and CYP3A4 were determined in vitro using HepG2 cells treated with different concentrations of Tween 80 (0.025, 0.05, and 0.1 mg/mL) or EL−35 (0.05, 0.1, and 0.2 mg/mL) for 24 h. Neither agent was cytotoxic to HepG2 cells at the examined concentrations (cell viability exceeding 90%) according to MTT assays (Appendix A). RT-qPCR and Western blotting demonstrated that the mRNA and protein expression of CYP2C8 and CYP3A4 was downregulated by EL−35 at concentrations of 0.1 and 0.2 mg/mL, whereas Tween 80 did not affect the expression of CYP2C8 and CYP3A4 at the tested concentrations (Figure 2 and Figure 3).

### 3.3. Effects of Tween 80 and EL−35 on the Pharmacokinetics of PTX in Wistar Rats after Single or Multiple Doses

To further understand the potential interaction of Tween 80 and EL−35 on CYP2C8-mediated metabolism in vivo, we determined the pharmacokinetics of PTX after single- or multiple-dose administration via caudal vein injection. No change in the plasma concentration–time curves of PTX were observed after single-dose administration of both Tween 80 and EL−35, as well as multiple-dose administration of Tween 80, whereas the elimination phase of the concentration–time curve of PTX was significantly elevated after multiple-dose administration of EL-35 (Figure 4). The pharmacokinetic parameters of PTX after single- or multiple-dose administration of the PEs, including half-life (t_1/2_), elimination rate constants (*k*), peak concentration (C_max_), apparent volume of distribution (Vd), area under the concentration–time curve (AUC), clearance (CL), and mean residence time (MRT), are presented in Table 1. No parameters were changed by single doses of either PE or by multiple doses of Tween 80, in line with the concentration–time curve. However, the AUC and MRT of PTX were significantly increased after multiple-dose administration of EL-35 compared with the findings for saline, whereas CL and *k* decreased.

Moreover, we monitored the serum indices of liver function at the end of the multiple-dose administration of PEs. AST, ALT, and ALP levels did not differ between PE administration and the saline control (Appendix A).

### 3.4. EL−35 Inhibited the Activities and Expression of CYP2C8 in Wistar Rats

To confirm whether the pharmacokinetic changes in PTX were attributable to the downregulation of Cyp2c22 (CYP2C8 in humans) by PEs, we detected the hepatic expression of Cyp2c22 after multiple-dose administration of PEs. In addition, we monitored the content of Cyp2c11 (the predominant isoform in the male rat liver), Cyp2c6 (the other major isoform of Cyp2c in the rat liver), and Cyp3a1/2 (CYP3A4 in humans) in this study to elucidate the mechanism by which the pharmacokinetics of PTX was altered by multiple-dose PE exposure. The results indicated that the mRNA expression of Cyp2c6, Cyp3a1, and Cyp3a2 was decreased after 14 days of Tween 80 or EL−35 administration, but neither PE altered the expression of Cyp2c11. Cyp2c22 was downregulated by multiple-dose EL−35 administration, but not by Tween 80 (Figure 5). Meanwhile, we extracted liver microsomes after multiple-dose administration of PEs for 14 days and assessed PTX 6α′-hydroxylation in vitro. The results indicated that 6-OH-PTX production was significantly decreased after multiple doses of EL−35, but not Tween 80 (Figure 6).

## 4. Discussion

In vitro metabolism studies illustrated that Tween 80 and EL−35 consistently inhibited PTX 6α′-hydroxylation in both HLMs and RLMs. The IC_50_s of Tween 80 and EL−35 indicated a similar inhibitory effect on CYP2C8 activity. The Lineweaver–Burk plot in the presence and absence of PEs provides a quick visual impression of the different forms of enzyme inhibition. For instance, competitive inhibitors have the same y-intercept but different slopes and x-intercepts between two datasets, noncompetitive inhibitors produce plots with the same x-intercept but different slopes and y-intercepts, and uncompetitive inhibitors produce a series of parallel lines with different intercepts on the y- and x-axes [22]. However, the plots of Tween 80 and EL−35 did not match these three classical inhibition types; thus, we speculated that they may have a mixed inhibition type. As CYP2C8 can metabolize approximately 5–8% of drugs [22], it may have multiple active sites for various substrates with different chemical structures. Meanwhile, Tween 80 and EL−35 are macromolecular compounds that can block multiple active sites of CYP2C8 proteins, thereby producing a mixed inhibition pattern in vitro. In addition, the surfactants can disrupt enzyme activity, as a previous study found that several surfactants could inhibit CYP3A4 activity [27].

The cell experiments indicated that EL−35 could decrease the mRNA and protein contents of CYP2C8 and CYP3A4 in HepG2 cells, whereas Tween 80 had no such effect. Tween 80 and EL−35 are both widely used in drug formulations. For example, Taxol contains 527 mg EL−35 in a 5 mL injection, and Tween 80 is the major excipient of docetaxel injection. These drugs are used to treat several types of cancer, including ovarian cancer, breast cancer, lung cancer, Kaposi sarcoma, cervical cancer, and pancreatic cancer. Thus, patients could be repeatedly exposed to Tween 80 and EL−35 in vivo during treatment, potentially affecting CYP2C8 activity and leading to a PE–drug interaction.

In the in vivo studies, multiple doses of EL−35 increased the AUC and MRT of PTX and decreased CL and *k*. In contrast, no change in the concentration–time curves and relative pharmacokinetic parameters of PTX were observed after single-dose administration of both Tween 80 and EL−35 as well as multiple-dose administration of Tween 80. Generally, PEs in drug formulations do not reach an effective concentration after a single dose. Thus, they cannot alter the pharmacokinetics of active compounds by interacting with CYP450s directly. Similarly, neither Tween 80 nor EL−35 altered the pharmacokinetics of PTX in rats after a single dose. However, after multiple doses, PEs could accumulate in cells and regulate CYP450s through one or more specific pathways, leading to changes in the pharmacokinetics of drugs. Meanwhile, multiple-dose administration of PEs did not affect Vd and liver function in rats. Thus, the changes in the PTX concentration–time curve and pharmacokinetic parameters may be attributable to changes in the expression of Cyp2c22.

To verify our conjecture, we detected the hepatic expression of Cyp2c22 and other primary Cyp2c/3a isoforms in the rat liver and assessed PTX 6α′-hydroxylation in liver microsomes after multiple-dose administration. The results indicated that multiple doses of both two PEs decreased the mRNA expression of Cyp2c6, Cyp3a1, and Cyp3a2. However, only EL−35 decreased the expression of Cyp2c22 and suppressed 6-OH-PTX production after 14 days of treatment. Therefore, it is highly possible that the downregulation of Cyp2c22 induced by EL−35 results in slower PTX metabolism in rats, higher drug exposure, and increased residence time.

Our study suggested that continuous treatments with EL−35 (430 mg/kg) decreased Cyp2c22 expression and led to the change in PTX exposure in rats. Similarly, this may occur in the clinical medication of Taxol, which contains 6 mg PTX and 527 mg EL−35 in each mL of solution. Patients who receive the treatment of Taxol may be exposed to 4.5~5.9 times the dose of EL−35 compared with the dosage set in this study (Appendix A. Estimation of EL−35 exposure in humans). Therefore, the interaction of EL−35 and PTX observed in our experiments could be clinically relevant. However, due to the lack of accurate quantitative methods for PEs, we cannot track PEs in vivo, and we do not know the exact concentration of PEs in the plasma or liver. Therefore, the results of cell and animal experiments are not sufficient to predict the potential interaction of PEs in the human body. As such, the effects of PEs in humans must be further verified by a randomized controlled clinical trial. Moreover, mathematical models that integrate in vitro findings with clinical pharmacokinetic data, such as a population pharmacokinetic model [26] or physiologically based pharmacokinetic model, can play an essential role in predicting the potential effects of an investigational PE on CYP450s in the human body [28,29].

## 5. Conclusions

Our study found that both Tween 80 and EL−35 can inhibit the activity of CYP2C8 in HLMs/RLMs. In addition, EL−35 suppressed CYP2C8 expression in HepG2 cells. Multiple doses of EL−35 reduced Cyp2c22 expression in the rat liver, leading to slower PTX metabolism, higher drug exposure, and prolonged residence times. The results can inform doctors or researchers to take PEs’ extra effects into consideration during drug formulation and administration.

## Figures and Tables

**Figure 1 pharmaceutics-13-01492-f001:**
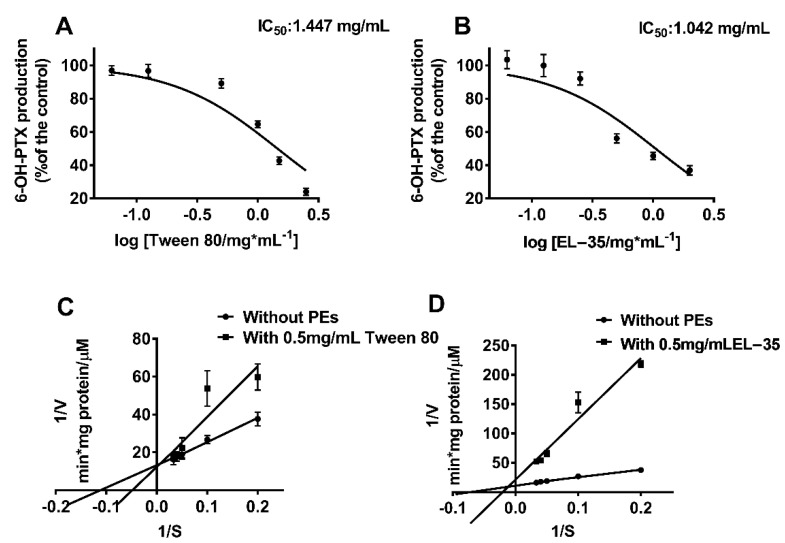
In vitro inhibition study of Tween 80 and EL−35 on CYP2C8 in HLM. (**A**,**B**) The IC_50_ determination of PEs on the inhibition of CYP2C8 activity. HLM was incubated with 10 μM PTX in the presence of a series of various concentrations of PEs for 1 h at 37 °C. The concentration range of each PE was as follows: EL−35 (0.03125–2 mg/mL), Tween 80 (0.0625–2.5 mg/mL). Activities are expressed as a percentage of the 6-OH-PTX production compared with the negative control. (**C**,**D**) Lineweaver–Burk for the inhibition of CYP2C8-mediated taxol 6-hydroxylation by various PEs (Tween 80, EL−35) with a concentration of 0.5 mg/mL in HLM. Each data point represents the mean of triplicate.

**Figure 2 pharmaceutics-13-01492-f002:**
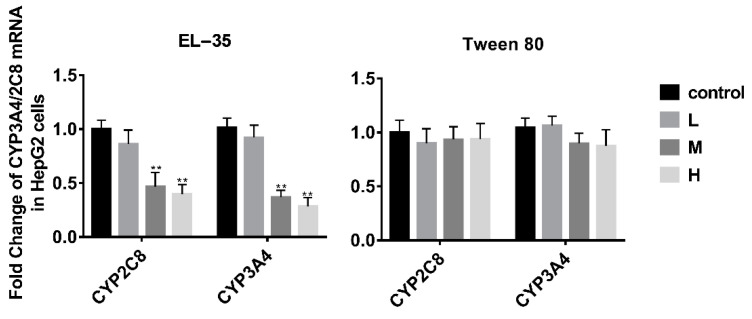
RT-qPCR analysis of the mRNA expression of CYP2C8 and CYP3A4 in HepG2 cells after treatment with different concentrations of Tween 80 and EL−35 for 24 h. The L/M/H concentrations were set as follows: 0.05/0.1/0.2 and 0.025/0.05/0.1 mg/mL for EL−35 and Tween 80, respectively. The mRNA expression levels of CYP2C8 and CYP3A4 were normalized to GAPDH. Data are expressed as the mean ± S.D. (*n* = 3 replicates/treatment). ** *p* < 0.01 against control.

**Figure 3 pharmaceutics-13-01492-f003:**
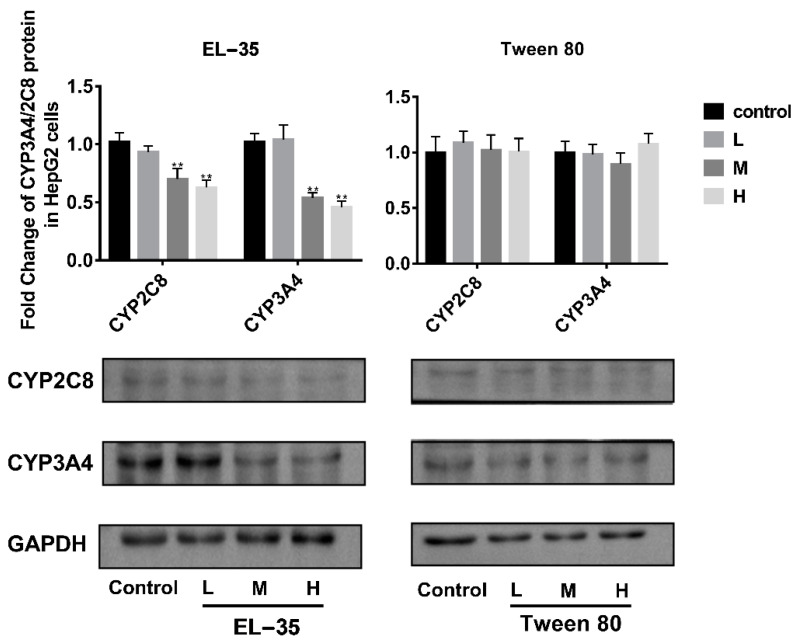
Western blot analysis of the protein expression of CYP2C8 and CYP3A4 in HepG2 cells after treatment with different concentrations of Tween 80 and EL−35 for 24 h. The L/M/H concentrations were set as follows: 0.05/0.1/0.2 and 0.025/0.05/0.1 mg/mL for EL−35 and Tween 80, respectively. The mRNA expression levels of CYP2C8 and CYP3A4 were normalized to GAPDH. Data are expressed as the mean ± S.D. (*n* = 3 replicates/treatment). ** *p* < 0.01, against control.

**Figure 4 pharmaceutics-13-01492-f004:**
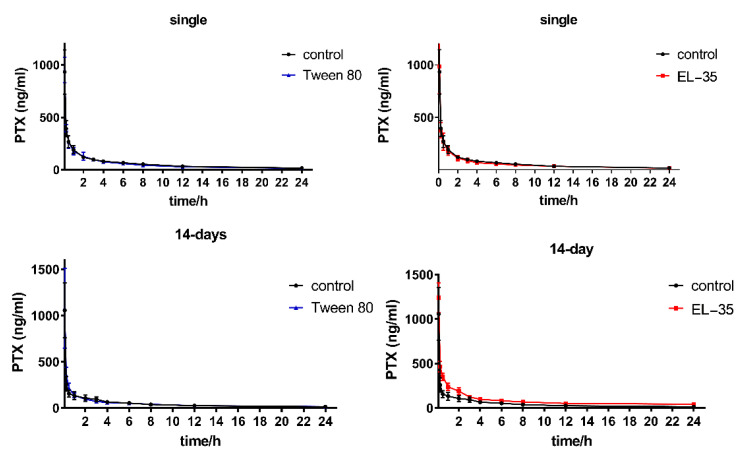
Plasma concentration–time plot of PTX after intravenous dosing presence and absence of Tween 80 or EL−35 with single-dose or multiple-dose administration (14 days), the time points were set as 6 min, 15 min, 30 min, 1 h, 2 h, 3 h, 4 h, 6 h, 8 h, 12 h, and 24 h.

**Figure 5 pharmaceutics-13-01492-f005:**
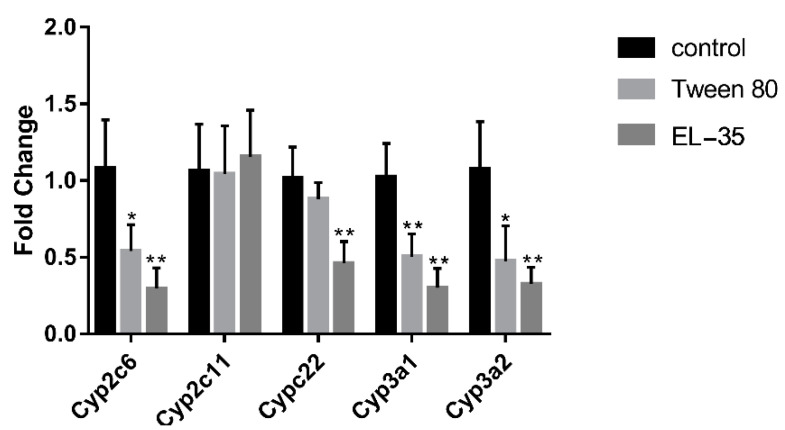
RT-qPCR analysis of the mRNA expression of Cyp2c and Cyp3a in Wistar rats’ liver after multiple administration of Tween 80 and EL−35 for 14 days. The mRNA expression levels of various Cyp2c and Cyp3a were normalized to Gapdh. Data are expressed as the mean ± S.D. (*n* = 6 replicates/treatment). * *p* < 0.05, ** *p* < 0.01, against control.

**Figure 6 pharmaceutics-13-01492-f006:**
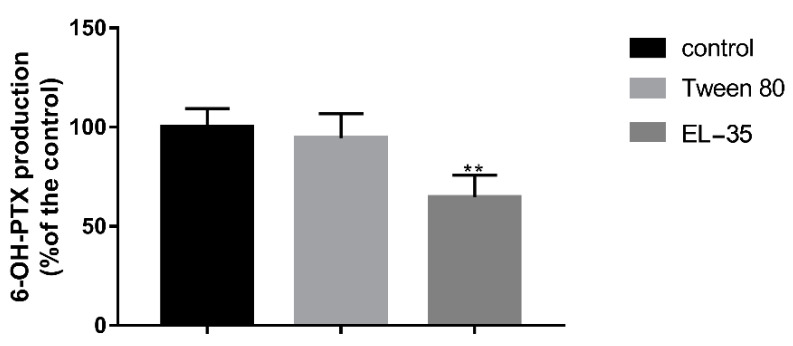
PTX-6α′ hydroxylation in rats’ liver microsomes after multiple-administration of PEs for 14 days. Data are expressed as the mean ± S.D. (*n* = 6 replicates/treatment). ** *p* < 0.01, against control.

**Table 1 pharmaceutics-13-01492-t001:** Summary of the pharmacokinetic parameters of PTX in the presence and absence of PEs with single/multiple-dose administration.

Compound, Dose, Route					PK Parameters, Mean ± SD, *n* = 6				
		t_1/2_	*k*	Cmax	Vd	AUC_(0-last)_	AUC_(0-inf)_	CL	MRT_(0-inf)_
		h	1/h	ng/mL	mL/kg	h × ng/mL	h × ng/mL	mL/h/kg	h
single-dose	PTX 3 mg/kg, iv + saline, iv	8.8 ± 0.9	0.08 ± 0.01	933.0 ± 237.1	16,682.8 ± 2797.0	1443.3 ± 133.9	1653.8 ± 160.4	1827.8 ± 178.6	10.1 ± 1.1
	PTX 3 mg/kg, iv + Tween 80, 180 mg/kg, iv	9.7 ± 3.1	0.08 ± 0.03	951.4 ± 134.6	18,030.3 ± 4788.1	1338.4 ± 257.3	1564.7 ± 368.4	1986.7 ± 370.5	10.4 ± 3.3
	PTX 3 mg/kg, iv + EL−35, 430 mg/kg, iv	10.0 ± 2.8	0.07 ± 0.02	985.4 ± 287.6	18,964.5 ± 5006.2	1338.8 ± 258.9	1574.0 ± 342.6	1995.7± 524.5	11.0 ± 3.4
14-days	PTX 3 mg/kg, iv + saline, iv	10.9 ± 4.6	0.07 ± 0.03	1057.0 ± 326.3	22,084.5 ± 8607.9	1146.4 ± 280.0	1379.4 ± 393.0	2313.8 ± 599.5	11.9 ± 4.3
	PTX 3 mg/kg, iv + Tween 80, 180 mg/kg, iv	11.2 ± 1.5	0.06 ± 0.01	1079.5 ± 471.1	22,407.0 ± 5218.8	1162.6 ± 223.9	1402.2 ± 276.8	2212.8 ± 447.1	11.9 ± 1.3
	PTX 3 mg/kg, iv + EL−35, 430 mg/kg, iv	20.4 ± 3.1 **	0.03 ± 0.01 **	1240.0 ± 181.2	21,207.4 ± 3102.1	2153.3 ± 316.6 **	3350.7 ± 674.4 **	923.4 ± 170.0 **	23.4 ± 4.0 **

** *p* < 0.01, against saline control.

## Data Availability

Not applicable.

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
