# Peer review of "Assessing CYP2C8-Mediated Pharmaceutical Excipient-Drug Interaction Potential: A Case Study of Tween 80 and Cremophor EL−35"

_pharmaceutics, 2021, doi:10.3390/pharmaceutics13091492_

Round 1

Reviewer 1 Report

Chengming Wen assessed the CYP2C8 mediated paclitaxel-excipient interactions using Tween 80 and Cremophor EL35. The manuscript is well written describing the paclitaxel-excipient interactions.

Below are the comments and provide the information in the manuscript:

  1. Line 92-The authors need to include the gender-related details for the HLM source.
  2. Mention the source information for heparin.
  3. The authors should include the blood volume collection and anesthesia procedure for animals and experimental design section.
  4. In the 2.4 section, what is the age of animals used for RLM extraction?
  5. LCMS chromatograms for bioanalytical samples need to be provided in the supplementary file. The authors should include the sample processing procedure for paclitaxel and its metabolite.
  6. In figure 1, results showed only for HLM data. The authors mentioned the RLM extraction procedure in the 2.4 section but didn’t provide the IC50 values, inhibition profiles, and Lineweaver Burk plots in RLM in figure 1. The authors should include the data in figure 1. The data symbols are very small and not clearly visible and distinguishable.
  7. HepG2 not “HepG2”. Modify this in the y-axes of figures 2 and 3.
  8. The authors developed the LCMS method for Paclitaxel metabolite but didn’t quantify the plasma concentrations with and without excipients. This is very important pharmacokinetic data for comparison of paclitaxel and its metabolite in plasma showing the effect of excipients on paclitaxel and its metabolite in figure 4.
  9. In figure 4, inset images should be included demonstrating the initial time points of PK data.
  10. In table 1, the volume of distribution and elimination rate constants data is needed. There was a big difference between AUC0-t and AUC0-inf for PTX and EL35 multiple-dose. What could be the reason for this major difference?
  11. Why the authors didn’t measure the liver concentrations of paclitaxel and its metabolite after single- and multi- dose administration of excipients? These results can be correlated with in vivo PK data.
  12. Genebank accession number/code should be included in Supplementary Table S2.

Author Response

Thank you for your review of our manuscripts. Those comments are constructive for us to improve the manuscript. Below are the responses for each comment:

  1. Line 92-The authors need to include the gender-related details for the HLM source.

Response:

The gender-related details of HLM have been added in “Materials” (line-95 red words)

  1. Mention the source information for heparin.

Response:

Source information of heparin has been added in “Materials” (line-94 red words)

  1. The authors should include the blood volume collection and anesthesia procedure for animals and experimental design section.

Response:

The blood volume collection and anesthesia procedure for animals have been added in “methods 2.2” (line-110 and line-113 red words)

  1. In the 2.4 section, what is the age of animals used for RLM extraction?

Response:

The age of animals used for RLM extraction has been added in “methods 2.4” (line-142 red words)

  1. LCMS chromatograms for bioanalytical samples need to be provided in the supplementary file. The authors should include the sample processing procedure for paclitaxel and its metabolite.

Response:

LCMS chromatograms for PTX and 6-OH-PTX in rats’ plasma and metabolism incubation system have been added in the supplementary file (Figure S5 and S6); sample processing procedure for paclitaxel has been added in “methods 2.5 (line-161~165 red words)”

  1. In figure 1, results showed only for HLM data. The authors mentioned the RLM extraction procedure in the 2.4 section but didn’t provide the IC50values, inhibition profiles, and Lineweaver Burk plots in RLM in figure 1. The authors should include the data in figure 1. The data symbols are very small and not clearly visible and distinguishable.

Response:

1) We have adjusted the scale of data symbols in Figure 1 to make them more visible and distinguishable.

2) About the question of why we extracted the RLM but didn’t provide the IC50 values, inhibition profiles, and Lineweaver Burk plots in RLM:

First, we do evaluate the inhibition effects of PEs in both HLM and RLM (data showed in supplementary file Figure S2),

Second, the IC50 values and Lineweaver Burk plots were used to preliminarily assessed the inhibition potential and inhibition types of these two PEs on CYP2C8 in this study. We believe that the evaluation results in the HLM system may have better meanings for predictions PE-Drug interaction potential in the human body based on mathematical models in subsequent studies. Besides, we have evaluated the effects of these two PEs on CYP2C8 mediated metabolism in rats in the follow-up study of this article. Therefore, we only fit the IC50 values and Lineweaver Burk plots in the in vitro HLM system, not in RLM.

  1. HepG2 not “HepG2”. Modify this in the y-axes of figures 2 and 3.

Response:

We have corrected the “HepG2” to “HepG2” in the y-axes of figures 2 and 3.

  1. The authors developed the LCMS method for Paclitaxel metabolite but didn’t quantify the plasma concentrations with and without excipients. This is very important pharmacokinetic data for comparison of paclitaxel and its metabolite in plasma showing the effect of excipients on paclitaxel and its metabolite in figure 4.

Response:

The LCMS method for 6-OH-PTX built in this study was mainly used to analyze its concentration in the in vitro metabolism system to reflect the activity of CYP2C8. We have tried to simultaneously analyze the concentration of PTX and 6-OH-PTX in rats’ plasma, but the concentration of 6-OH-PTX in plasma samples was much lower than the LLOQ of the built method. Instead, we extracted the liver microsome of rats after multiple-dose administration of PEs and determined the production of 6-OH-PTX in vitro to reflect the change of CYP2C8 mediated metabolism in rats after multiple-dose of PEs.

Besides, in the study of Giri P and his colleagues that assessed the inhibition effects of saroglitazar magnesium on CYP2C8 in rats using several substances of CYP2C8 include PTX, the authors only quantified the plasma concentration of PTX in the study too. (Giri P et al. Eur J Pharm Sci. 2019 Mar 15;130:107-113)

  1. In figure 4, inset images should be included demonstrating the initial time points of PK data.

Response:

The time points if PK data have been added to the figure legend of Figure 4.

  1. In table 1, the volume of distribution and elimination rate constants data is needed. There was a big difference between AUC0-t and AUC0-inf for PTX and EL35 multiple-dose. What could be the reason for this major difference?

Response:

We have added the data of the volume of distribution and elimination rate constants in Table 1. Multiple-dose administration of EL-35 didn’t affect the volume of distribution, but decreased the elimination rate (in line with the changes of t1/2 and CL), together with all the results presented in “Results section 3.3 and 3.4”, we believe that the reason for the major difference between AUC0-t and AUC0-inf for PTX and EL35 multiple-dose should be the downregulation of Cyp2c22 induced by EL-35 that results in slower PTX metabolism in rats.

  1. Why the authors didn’t measure the liver concentrations of paclitaxel and its metabolite after single- and multi- dose administration of excipients? These results can be correlated with in vivo PK data.

Response:

The liver concentrations of paclitaxel and its metabolite may reflect the effects of compounds on the distribution of PTX. The PK study in rats demonstrated that PEs (assessed in this study) didn’t affect the volume of distribution of PTX with single- or multiple-dose, which suggested that PEs may have limited influence on the distribution of PTX. Besides, in this study, we mainly focus on the effects of PEs on CYP2C8 mediated metabolism. Therefore, we didn’t measure the liver concentrations of paclitaxel and its metabolite after single- and multi- dose administration of excipients.

Furthermore, according to the published study on the effects of compounds on CYP450s mediated metabolism in rats, most researchers didn’t measure the liver concentrations of substrate and its metabolite. (Giri, P et al. Eur J Pharm Sci 2019, 130, 107-113; Chen J et al. Pharmazie. 2020 Sep 1;75(9):424-429; Wang G et al. Pharmacology. 2020;105(1-2):79-89; Zayed A et al. 2020 Jun 19;12:169-179; etc.)

  1. Genebank accession number/code should be included in Supplementary Table S2.

Response:

Genebank accession numbers of all the primers have been added in Supplementary Table S2.

Reviewer 2 Report

The paper from Wen et al. assesses the impact of Cremophor EL and Tween 80 on the PK of paclitaxel (Taxol formulation). The interesting topic is well addressed, adequately investigated and the conclusions do sound. To my best knowledge the paper is new and unique. The paper could even increase in quality by performing the following amendments: 

  • Major issues:
    - The following paper is missing in the introduction and in the discussion and must be adequately included: 
    Fransson MN et al. (2011). Influence of Cremophor EL and genetic polymorphisms on the pharmacokinetics of paclitaxel and its metabolites using a mechanism-based model. Drug Metab. Dis. 39 (2) 247-255.
    - Also Bergmann et al. (2011) Pharmacogen J could be helpful.
    - Provide the ethical commission which gave consent and -if applicable- the study code for the rodent studies.
  • Minor issues:
    - Abstract: PEs may also affect drug carriers and efflux transporters (l.12). Do not use PTX as abbreviation in the abstract or at least introduce it once (l.15).
    - Introduction: there are much more functionalities of excipients (l.31), also for Tween 80 and Cremophor EL in addition to "surfactants" (emulsifiers, wetting agent, solubilizers etc.).
    - Materials, l. 87: write "Macrogol 600 (PEG 600)" as macrogol is the compendial term).
    - Results and discussion: try to calculate / estimate the meaning of the interaction of Cremophor EL and paclitaxel observed in your experiments. Does the magnitude of interaction and doses reflect the clinical observations in literature?  Or is it x-fold lower?

Author Response

Thank you for your review of our manuscripts. Those comments are constructive for us to improve the manuscript. Below are the responses for each comment:

  • Major issues:
    - The following paper is missing in the introduction and in the discussion and must be adequately included: 
    Fransson MN et al. (2011). Influence of Cremophor EL and genetic polymorphisms on the pharmacokinetics of paclitaxel and its metabolites using a mechanism-based model. Drug Metab. Dis. 39 (2) 247-255.
    - Also Bergmann et al. (2011) Pharmacogen J could be helpful.
    - Provide the ethical commission which gave consent and -if applicable- the study code for the rodent studies.

Response:

1) We have added relative descriptions in the introduction and discussion section according to the paper "Fransson MN et al. Drug Metab. Dis. 39 (2) 247-255." (line-79~80 and line-373 red words)

However, we can't find the article "Bergmann et al. (2011) Pharmacogen J", so we didn't add it into the manuscript.

2) The ethical commission and study code for the rodent studies were list at Institutional Review Board Statement (line-396~398 red words).

  • Minor issues:
    - Abstract: PEs may also affect drug carriers and efflux transporters (l.12). Do not use PTX as abbreviation in the abstract or at least introduce it once (l.15).
    - Introduction: there are much more functionalities of excipients (l.31), also for Tween 80 and Cremophor EL in addition to "surfactants" (emulsifiers, wetting agent, solubilizers etc.).
    - Materials, l. 87: write "Macrogol 600 (PEG 600)" as macrogol is the compendial term).

Response:

We have corrected all these issues in our manuscript at line-12, -15, -33, and -89 (red words)

- Results and discussion: try to calculate / estimate the meaning of the interaction of Cremophor EL and paclitaxel observed in your experiments. Does the magnitude of interaction and doses reflect the clinical observations in literature?  Or is it x-fold lower?

Response:

We have estimated the exposure of EL-35 in human during the clinical medication of Taxol and compared with the dose set in this study (data was list in Supplemental file 3. Estimation of EL-35 exposure in human). We found that patients who receive the treatment of Taxol may expose to 4.5~5.9 times the dose of EL-35 compared with the dosage set in this study. Therefore, the interaction of EL-35 and PTX observed in our experiments could be clinically relevant. The related description has been added to the "Discussions" (line-361~367 red words).

Round 2

Reviewer 1 Report

The authors significantly improved the manuscript and answered all the comments.